# Unprecedented yet gradual nature of first millennium CE intercontinental crop plant dispersal revealed in ancient Negev desert refuse

Daniel Fuks[1,2]*, Yoel Melamed[3], Dafna Langgut[4], Tali Erickson-Gini[5], Yotam Tepper[6,7], Guy Bar-Oz[7], Ehud Weiss[3]

[1]McDonald Institute for Archaeological Research, Department of Archaeology, University of Cambridge, Cambridge, United Kingdom; [2]Department of Archaeology, Ben-Gurion University of the Negev, Beer-Sheva, Israel; [3]Archaeobotany Lab, Martin (Szusz) Department of Land of Israel Studies and Archaeology, Bar-Ilan University, Ramat-Gan, Israel; [4]Laboratory of Archaeobotany and Ancient Environments, Institute of Archaeology & The Steinhardt Museum of Natural History, Tel Aviv University, Tel Aviv, Israel; [5]Southern Region, Israel Antiquities Authority, Omer Industrial Park, Omer, Israel; [6]Central Region, Israel Antiquities Authority, HiPort, Shoham, Israel; [7]School of Archaeology and Maritime Cultures, University of Haifa, Haifa, Israel

*For correspondence:
df427@cam.ac.uk; dfuks@bgu.ac.il

Competing interest: The authors declare that no competing interests exist.

**Abstract** Global agro-biodiversity has resulted from processes of plant migration and agricultural adoption. Although critically affecting current diversity, crop diffusion from Classical antiquity to the Middle Ages is poorly researched, overshadowed by studies on that of prehistoric periods. A new archaeobotanical dataset from three Negev Highland desert sites demonstrates the first millennium CE's significance for long-term agricultural change in Southwest Asia. This enables evaluation of the 'Islamic Green Revolution (IGR)' thesis compared to 'Roman Agricultural Diffusion (RAD)', and both versus crop diffusion during and since the Neolithic. Among the findings, some of the earliest aubergine (*Solanum melongena*) seeds in the Levant represent the proposed IGR. Several other identified economic plants, including two unprecedented in Levantine archaeobotany—jujube (*Ziziphus jujuba/mauritiana*) and white lupine (*Lupinus albus*)—implicate RAD as the greater force for crop migrations. Altogether the evidence supports a gradualist model for Holocene-wide crop diffusion, within which the first millennium CE contributed more to global agricultural diversity than any earlier period.

## Editor's evaluation

The study presents important findings on the timing and movement of crops in the Near East. The authors provide convincing data supporting a predominant contribution of Roman Agricultural Diffusion to the spread of a number of cultigens in the region. The work will be of interest to those thinking about the timing and movement of the diffusion of agricultural crops post-domestication.

## Introduction

Crop diversity has long been recognized as key to sustainable agriculture and global food security, encompassing genetic resources for agricultural crop improvement geared at improving yields,

pest resistance, climate change resilience, and the promotion of cultural heritage. The current global genetic diversity of agricultural crops is a product of their dispersal from multiple regions and much research has attempted to reconstruct these trajectories (*Vavilov, 2009*; *Diamond, 2002*; *Zohary et al., 2012*). As part of this effort, archaeobotanical research on plant migrations across the Eurasian continent has been a central theme in recent decades, especially with reference to 'food globalization' and the 'Trans-Eurasian exchange' (*Jones et al., 2011*; *Boivin et al., 2012*; *Liu et al., 2019*; *Sherratt, 2006*; *Zhou et al., 2020*). Yet, as is true for archaeology-based domestication research in general, most studies of crop dispersal and exchange have focused on prehistoric origins and developments (*Zohary and Hopf, 1973*; *Smith, 1989*; *Denham et al., 2003*; *Tanno and Willcox, 2006*; *Weiss et al., 2006*; *Purugganan and Fuller, 2009*; *Riehl et al., 2013*), to the near exclusion of more recent crop histories directly affecting today's agricultural diversity. One of the most influential, and contested, chapters in the later history of crop diffusion is the 'Islamic Green Revolution' (IGR) (*Watson, 1983*; *Decker, 2009a*; *Squatriti, 2014*). According to Andrew Watson, the IGR involved a package of sub-/tropical, mostly East- and South Asian domesticates which, as a result of Islamicate territorial expansion, spread into Mediterranean lands along with requisite irrigation technologies ca. 700–1100 CE. This allegedly involved some 17 domesticated plant taxa (*Table 1*), including such economically significant crops as sugar cane, orange, and banana (*Watson, 1983*). However, critics have argued that many of the proposed IGR crops were, and still are, of minor economic significance, while others were previously cultivated in the Mediterranean region, particularly under Roman rule, or else arrived much later (*Decker, 2009a*; *Johns, 1984*; *Ashtor, 1985*). Indeed, there is considerable evidence for crop diffusion immediately preceding and during the Roman period in the Eastern Mediterranean, 1st c. BCE–4th c. CE. During this time, several East- and Central Asian crops, including some of those on Watson's IGR list (e.g. lemon, rice), appear to have been first introduced to the Mediterranean region, along with agricultural technologies (*Decker, 2009a*; *Johns, 1984*; *Ashtor, 1985*; *Decker, 2009b*; *Kamash, 2012*). From this period on, a growing fruit basket is evident in sites and texts of the Eastern Mediterranean region (*Amar, 2000*; *Kislev and Simchoni, 2006*; *Aubaile, 2012*; *Amichay et al., 2019*). These include several tree fruits such as peach, pear, plum, hazel, and others (*Table 2*) apparently reflecting the Greco-Roman passion for grafting and its pivotal role in the dispersal of mostly temperate fruit crops from Central Asia to the Mediterranean and Europe (*Zohary et al., 2012*;

**Table 1.** Proposed Islamic Green Revolution (IGR) crops (according to *Watson, 1983*).

| Category | Latin name | English common name |
| --- | --- | --- |
| cereal | *Sorghum bicolor* (L.) Moench. | sorghum |
| | *Oryza sativa* L. | rice |
| | *Triticum durum* Desf. | hard wheat |
| tree fruit | *Citrus aurantium* L. | sour orange |
| | *Citrus limon* (L.) Osbeck | lemon |
| | *Citrus aurantiifolia* (Christm.) Swingle | lime |
| | *Citrus maxima* (Burm.) Merr. | shaddock |
| | *Musa paradisiaca* L. | banana/plantain |
| | *Cocos nucifera* L. | coconut |
| | *Mangifera indica* L. | mango |
| vegetable | *Citrullus lanatus* (Thunb.) Matsum. & Nakai | watermelon |
| | *Spinacia oleracea* L. | spinach |
| | *Cynara cardunculus* L. | artichoke |
| | *Colocasia esculenta* (L.) Schott | taro |
| | *Solanum melongena* L. | eggplant |
| condiment | *Saccharum officinarum* L. | sugar cane |
| textile | *Gossypium arboreum/herbaceum* L. | Old World cotton |

**Table 2.** Proposed Roman Agricultural Diffusion (RAD) crops in the Eastern Mediterranean*.

| Category | Latin name | English common name |
|---|---|---|
| cereal | *Oryza sativa* L. | rice |
| | *Sorghum bicolor* (L.) Moench. | sorghum |
| legume | *Lupinus albus* L. | white lupine |
| tree fruit/nut | *Ceratonia siliqua* L. | carob |
| | *Morus nigra* L. | black mulberry |
| | *Prunus persica* (L.) Batsch | peach |
| | *Pyrus communis* L. | pear |
| | *Prunus domestica* L. | plum |
| | *Prunus armeniaca* L. | apricot |
| | *Prunus avium/cerasus* L. | cherry |
| | *Pistacia vera* L. | pistachio nut |
| | *Pinus pinea* L. | stone pine |
| | *Corylus* sp. | hazel |
| | *Ziziphus jujuba/mauritiana* | jujube |
| | *Citrus limon* (L.) Osbeck | lemon |
| | *Cocos nucifera* L. | coconut |
| vegetable | *Cucumis melo* convar. *melo* | muskmelon |
| textile | *Cannabis sativa* L. | hemp |

*Includes species first attested in the 1st c. BCE Hellenistic-Roman transition. Although carob is a native Mediterranean tree, improved food cultivars are first attested in this period. Similarly, stone pine is first attested during this period in the S Levant, although native to the NE Mediterranean.

*Mudge et al., 2009*). Yet Roman arboricultural diffusion is but a subset of Roman agricultural diffusion (hereafter, RAD), which also includes non-arboreal crops (including hemp, muskmelon, white lupine, rice, sorghum) and various agricultural techniques diffused by the Romans into the Eastern Mediterranean (*Kamash, 2012*; *Mercuri et al., 2002*; *Pelling, 2005*; *Cappers, 2006*; *Van der Veen, 2011*; *Wilson, 2002*; *Kron, 2012*; *Avital, 2014*; *Van der Veen et al., 2008*; *Butzer et al., 1985*). Not all crops in motion during this period took hold in local agriculture. In some cases, as has been claimed for rice in Egypt, Roman-period importation of the new crops was followed by local cultivation in the Islamic period (*Van der Veen et al., 2018*). In other cases, Roman introductions were subsequently abandoned (*Livarda, 2011*), or failed to diffuse beyond elite gardens until much later (*Langgut, 2017*). Limited adoption in local agriculture is also a feature of some proposed IGR crops, as Watson admitted regarding coconut and mango (*Watson, 1983*). Thus, a cursory consideration of proposed IGR and RAD crops in the Eastern Mediterranean reveals that the balance between the two is about even and perhaps weighted toward RAD (*Tables 1–2*). This sort of comparison is useful for evaluating the IGR thesis and attaining improved understandings of crop dispersal in the first millennium CE, but a higher-resolution micro-regional approach is needed to rigorously gauge these developments. Systematic evaluation of respective Islamic and Roman contributions to agricultural dispersal has been attempted for Iberia (*Butzer et al., 1985*; *Peña-Chocarro et al., 2019*). In the Eastern Mediterranean, archaeobotanical studies in Egypt (*Van der Veen et al., 2018*), northern Syria (*Samuel, 2001*), and Jerusalem (*Amichay et al., 2019*; *Fuks et al., 2020a*) have also yielded evidence for IGR introductions framed against Roman agricultural diffusion, but these have not yet been considered holistically.

The exceedingly rich plant remains from relatively undisturbed Negev Highland middens (*Figures 1–2*; *Bar-Oz et al., 2019*; *Tepper et al., 2018a*; *Tepper et al., 2020a*) provide a significant new addition to the evidence for Levantine and Mediterranean crop diffusion, informing upon changes in the local economic plant basket over the first millennium CE. The Negev Highlands offer an

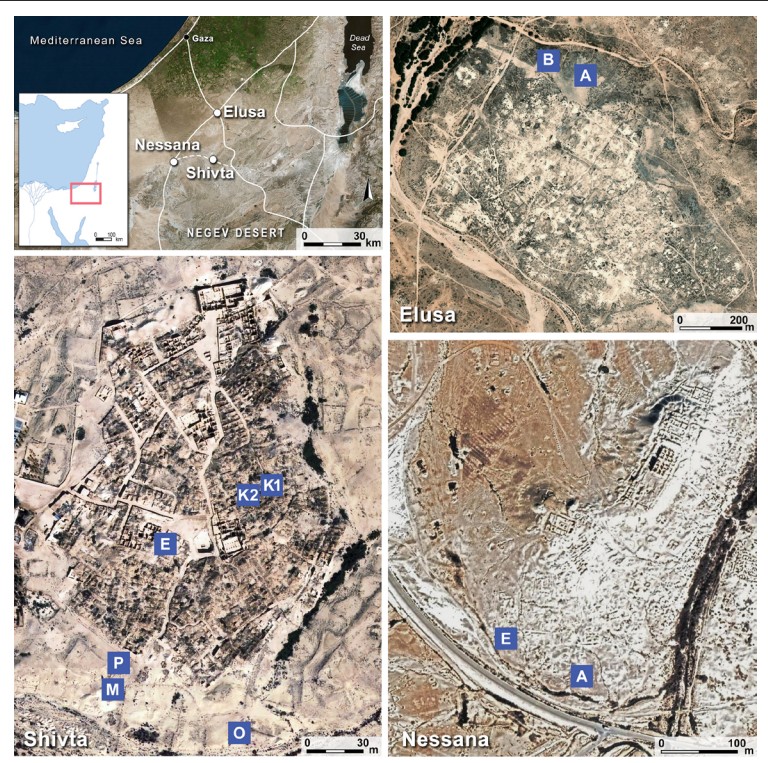

**Figure 1.** Study sites and middens. The study sites—Shivta, Elusa, and Nessana—roughly span the Negev Highlands region of the Negev desert. The excavated middens are marked on the aerial photos above and are lettered as named in the 2015–2017 excavations (see also *Table 4*). Adjacent middens include K1-K2 at Shivta and A1-A4 at Elusa; the latter are marked above only as A.

ideal test case for the geographical extent of crop dispersal, as a desert region on the margins of the settled zone, which practiced vibrant runoff farming and engaged in Mediterranean–Red Sea trade (*Kedar, 1957*; *Evenari et al., 1982*; *Tepper et al., 2020b*; *Bruins et al., 2019*; *Fuks et al., 2021a*). Archaeobotanical finds from the Negev Highlands, mainly from Byzantine sites (4th–7th centuries CE), have been reported in previous studies (*Bar-Oz et al., 2019*; *Tepper et al., 2018a*; *Mayerson, 1962*; *Liphschitz, 2004*; *Ramsay et al., 2016*; *Fuks et al., 2020b*; *Tepper et al., 2018b*; *Fuks et al., 2016*; *Dunseth et al., 2019*; *Fuks and Dunseth, 2021b*; *Langgut et al., 2021*), including those deriving from organically rich middens at Elusa, Shivta, and Nessana, excavated as part of the recent NEGEVBYZ project (*Fuks et al., 2020b*; *Tepper et al., 2018b*; *Fuks et al., 2016*; *Dunseth et al., 2019*; *Fuks and Dunseth, 2021b*; *Langgut et al., 2021*). We present below the first complete dataset of identified plant remains from the Late Antique Negev Highland middens dated to the local Roman, Byzantine, and Early Islamic periods (2nd–8th centuries CE). We then analyze this data to assess the evidence for Roman and Early Islamic crop diffusion in the Southern Levant, comparing with earlier introductions. These include the Southwest Asian Neolithic 'founder crops,' Chalcolithic-Early Bronze Age fruit tree domesticates, and Bronze-Iron Age introductions (*Tables 1–3*). This analysis offers Holocene-scale insights into the dynamics of crop diffusion.

## Results

Roughly 50,000 quantifiable macroscopic plant parts were retrieved from fine-sifted flotation and dry-sieved sediment samples of the middens of Elusa, Shivta, and Nessana, excluding charcoal and in addition to a roughly equal number retrieved from wet-sieving. These mostly seed and fruit (carpological) remains were identified to a total of 144 distinct plant taxa (*Supplementary file 1*). Nearly half of the identified specimens were derived from six Shivta middens, one quarter from three Elusa middens, and one quarter from two Nessana middens. Preservation quality varied somewhat within

**Table 3.** Pre-1st millennium CE introductions/domestications in the Southern Levant*.

| Period | Category | Latin name | English common name |
|---|---|---|---|
| Neolithic | cereal | *Hordeum vulgare* subsp. *vulgare* | barley |
| | | *Triticum monococcum* subsp. *monococcum* | einkorn wheat |
| | | *T. turgidum* subsp. *dicoccum* (Schrank ex Schübl.) Thell. | emmer wheat |
| | | *T. aestivum/durum* s.l. | free-threshing wheats |
| | legume | *Lens culinaris* Medik. syn. *Vicia lens* (L.) Coss. & Germ. | lentil |
| | | *Pisum sativum* L. syn. *Lathyrus oleraceus* Lam. | pea |
| | | *Cicer arietinum* L. | chickpea |
| | | *Vicia ervilia* (L.) Willd. | bitter vetch |
| | | *Vicia faba* L. | broad bean |
| | fiber/oil | *Linum usitatissimum* L. | flax |
| Chalcolithic | tree fruit/nut | *Olea europaea* L. | olive |
| | | *Vitis vinifera* L. | grapevine |
| | | *Ficus carica* L. | fig |
| | | *Ficus sycomorus* L. | sycomore fig |
| | | *Phoenix dactylifera* L. | date |
| | | *Punica granatum* L. | pomegranate |
| | | *Prunus amygdalus* Batsch | almond |
| Bronze-Iron Age | cereal | *Panicum miliaceum* L. | broomcorn millet |
| | legume | *Lathyrus clymenum* L. | Spanish vetchling |
| | | *Lathyrus sativus/cicera* L. | grass/red pea |
| | | *Trigonella foenum-graecum* L. | fenugreek |
| | tree fruit/nut | *Juglans regia* L. | walnut |
| | | *Citrus medica* L. | citron |
| | vegetable | *Citrullus lanatus* (Thunb.) Matsum. & Nakai | watermelon |
| | condiment/oil | *Papaver somniferum* L. | opium poppy |
| | | *Nigella sativa* L. | black cumin |
| | | *Sesamum indicum* L. | sesame |

*Based primarily on **Zohary et al., 2012**, this list includes only species whose evidence for domestication/introduction is clear. This and the preceding tables are not intended to be exhaustive lists but rather to provide a basis against which the Negev Highlands crop plant assemblage can be compared.

and between middens and samples, the richest of which were the Early Islamic middens from Shivta and Nessana, which also displayed a higher diversity of finds (*Supplementary file 2*). However, all middens yielded rich concentrations of charred seeds and other organic remains, including many exceptionally preserved specimens. Identified species were classified as either domesticated or wild and the former were grouped by functional category (*Supplementary file 1*). Most of the 120 wild taxa have ethnographically documented uses, whether for forage or fodder, crafts or fuel, food or spice, medicine or recreation. Nearly all of them grow wild in the Negev Highlands today and we cannot determine for certain which were deliberately used on site. Twenty-three domesticated food plant taxa were identified by carpological remains, including cereals, legumes, fruits, nuts, and one vegetable. We focus on these plants as indicators of local foodways and global crop diffusion. Their orders of magnitude by midden context appear in *Table 2*, for specimens retrieved from fine-sifted samples (see *Materials and methods* for sampling strategy). This data enables categorization of Late

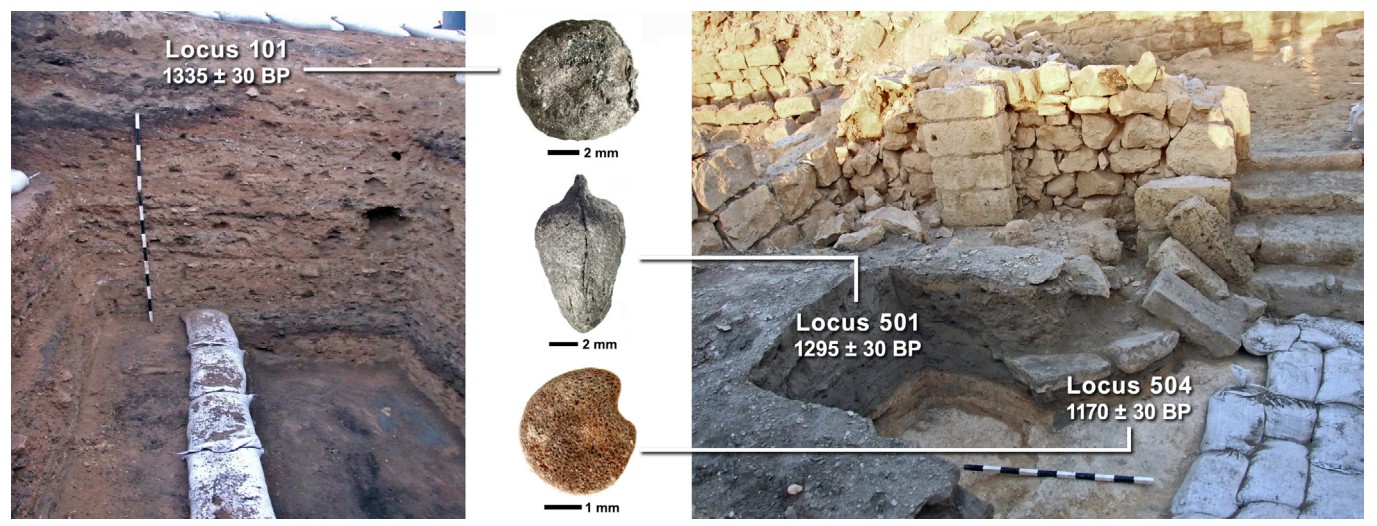

**Figure 2.** First finds from the Negev Highlands middens. Section photos of Nessana midden A (left) and Shivta midden E (right) are shown with select loci (photographed by Yotam Tepper) and their uncalibrated radiocarbon dates, from which remains of white lupine (center top), jujube (center middle), and aubergine (center bottom) were found. These plant remains represent some of the earliest of their species found in the Southern Levant (photographed by Daniel Fuks).

Antique Negev Highland domesticates as staples, cash crops, and luxury/supplementary foods, setting the stage for analysis of the local manifestation of long-term crop diffusion.

Identified charcoal and pollen previously reported by *Langgut et al., 2021* (*Supplementary files 3–5*) raise the number of distinct plant taxa identified in the NEGEVBYZ project to over 180. Among them, pollen of the exotic hazel (*Corylus* sp.)—apparently grown locally for its nuts—is included in the discussion of domesticated food plants. Doum palm (*Hyphaene thebaica* [L.] Mart.), which grows wild today in the southern Aravah valley, is attested by charcoal (*Supplementary files 3–5*) but this likely represents wild rather than domesticated specimens. Similarly, sycomore fig (*Ficus sycomorus* L.), which produces tasty fruits, was grown primarily for wood in ancient times (*Feliks, 1968*). Therefore, we exclude doum palm and sycomore fig from the discussion of domesticated food plants' status and longer-term trajectories, but include them among the fruit trees in *Supplementary file 5*.

Seed quantities and ubiquity point to barley (*Hordeum vulgare* L.), wheat (*Triticum turgidum/ aestivum*), and grape (*Vitis vinifera* L.) as the main cultivated crops, which were clearly calorific staples. Their local cultivation is attested to by cereal processing waste (rachis fragments, awn and glume fragments, culm nodes, and rhizomes) and wine-pressing waste (grape pips, skins, and pedicels). In addition, lentils (*Lens culinaris* Medik. syn. *Vicia lens* [L.] Coss. & Germ.), bitter vetch (*Vicia ervilia* [L.] Willd.), fig (*Ficus carica* L.), date (*Phoenix dactylifera* L.), and olive (*Olea europaea* L.) should also be counted as staples based on seed quantities and ubiquity (*Table 4*; *Supplementary file 5*). They too were likely cultivated locally. Significantly, all identified staples were among the Southwest Asian Neolithic founder crops and early fruit domesticates which formed a stable part of Levantine diets by the end of the Chalcolithic (c. 4500–3300 BCE).

Grapes were previously shown to be the primary cash crop of the Byzantine Negev Highlands, particularly in the mid-5th to mid-6th c. CE, based on their changing relative frequencies (*Fuks et al., 2020b*). Interestingly, free-threshing hexaploid bread wheat (*Triticum aestivum* s.l.)—a more market-oriented wheat species identifiable archaeologically by indicative rachis segments—appears in the Negev Highlands only after the mid-6th c. (*Table 4*). This corresponds with the period of decline in viticulture (*Fuks et al., 2020b*).

In the 'luxuries and supplements' category, we include potentially important and desirable dietary components which were minor and apparently nonessential in local consumption or agriculture. These include several food crops poorly represented in the local assemblages: broad bean (*Vicia faba* L.), fenugreek (*Trigonella foenum-graecum* L.), Spanish vetchling (*Lathyrus clymenum* L.), and white lupine (*Lupinus albus* L.) among the legumes; peach (*Prunus persica* [L.] Batsch), plum/cherry (*Prunus* subgen. *Cerasus/Prunus*), carob (*Ceratonia siliqua* L.), and jujube (*Ziziphus jujuba/mauritiana*) among

**Table 4.** Domesticated plant seeds in order of magnitude by period, site, and area (from fine-sift).

| Plant species | Century CE | 1st–3rd | 4th–mid-5th | mid-5th–mid-6th | | mid-5th–mid-7th | mid-6th–mid-7th | | early 7th | | 7th–8th | | mid-7th–8th | | |
|---|---|---|---|---|---|---|---|---|---|---|---|---|---|---|---|
| | Site | SVT | HLZ | HLZ | SVT | NZN | NZN | SVT | SVT | | NZN | NZN | SVT | | |
| | Area (midden) | P | A4 | A1 | M | A | A | O | K2 | E | A | E | K1 | K2 | E |
| | Samples | 5 | 14 | 19 | 14 | 7 | 5 | 12 | 3 | 3 | 27 | 10 | 13 | 13 | 12 |
| | Vol. (L) | 15 | 85 | 85 | 42 | 21 | 15 | 36 | 9 | 9 | 84 | 33 | 39 | 39 | 36 |
| **Plant species** | **Common name** | | | | | | | | | | | | | | |
| *Hordeum vulgare* | Barley | XX | XXX | XXX | XX | XXX | XX | XX | XXX | XXX | XXX | XXX | XXX | XXX | XXX |
| *Triticum* spp. | Wheat | XX | XX | XX | XX | X | X | X | XX | XX | XX | XXX | XXX | XXX | XXX |
| *Lens culinaris* | Lentil | | XX | XX | X | XX | | X | X | X | X | XX | XX | X | X |
| *Vicia ervilia* | Bitter vetch | X | X | X | X | X | X | X | X | XX | X | XX | XX | X | XX |
| *Trigonella foenum-graecum* | Fenugreek | | X | | | | | | | X | X | X | X | X | |
| *Lathyrus clymenum* | Spanish vetchling | | | | | | | | | | X | X | | | |
| *Lupinus albus* | White lupine | | | | | | | | | | | X | | | |
| *Vitis vinifera* | Grape | X | XX | XX | XX | XX | X | XX | XX | X | XXX | XXX | XXX | XXX | XX |
| *Ficus carica* | Fig | X | XXX | XXX | XX | X | X | XX | X | X | XX | X | X | XX | |
| *Olea europaea* | Olive | | X | | X | X | X | X | X | | X | XX | X | X | X |
| *Phoenix dactylifera* | Date | X | X | X | X | X | | X | X | X | | XX | XX | X | |
| *Punica granatum* | Pomegranate | | rind | | rind | X | rind | X | rind | | X | XX | X | X | X |
| *Ceratonia siliqua* | Carob | | | | | | | | | | X | X | pistil | | |
| *Prunus amygdalus* | Almond | | | | | | | | | | X | X | X | X | |
| *Prunus persica* | Peach | | X | | | | X | | | | X | X | | | |
| *Pinus pinea* | Stone pine | | | | | | | | | | X | X | | | |
| *Solanum melongena* | Aubergine | | | | | | | | | | X | | | | X |
| *Vachellia nilotica** | Nile acacia | | | X | X | X | | | | | | | | | |

SVT-Shivta; HLZ-Elusa; NZN-Nessana; for midden locations see **Figure 1**. Orders of magnitude presented as 1≤X<10≤XX<100≤XXX<1000. See Materials and Methods for sampling strategy. This table is based on source data in **Table 4—source data 1**, **Table 4—source data 2** and **Table 4—source data 3**.

*Although not necessarily a domesticate, we take this Egyptian wild plant to have been cultivated or imported into the Negev Highlands, as explained in the text.

The online version of this article includes the following source data for table 4:

**Source data 1.** Elusa carpological remains.

**Source data 2.** Shivta carpological remains.

**Source data 3.** Nessana carpological remains.

the tree-fruits; almond (*Prunus amygdalus* Batsch), walnut (*Juglans regia* L.), stone pine (*Pinus pinea* L.), pistachio (*Pistacia vera* L.) and hazel (*Corylus* sp.) among the nuts; aubergine (*Solanum melongena* L.) as a unique summer vegetable (**Figures 2 and 3**); and supplementary wild edibles such as beet (*Beta vulgaris* L.), coriander (*Coriandrum sativum* L.), and European bishop (*Bifora testiculata* [L.] Spreng.) (**Supplementary file 1**). The latter three grow wild in Israel today mostly north of the Negev Highlands; we count them as wild considering their small quantities and nearby distribution. Any of the above could have been cultivated in Negev Highland runoff farming or on site (**Evenari et al., 1982**; **Langgut et al., 2021**; **Tepper et al., 2022**).

Complementing the seed/fruit remains presented above, palynological and anthracological analyses support local cultivation of grapevine, fig, olive, date, pomegranate, carob, and the *Prunus* genus, which includes almond, peach, and/or plum/cherry (**Langgut et al., 2021**). Based on stone pine seed coats (**Figure 3d**), and the identification of Pinaceae pollen indicative of a pine other than the local Aleppo pine (*Pinus halepensis* Mill.), it is plausible that stone pine was cultivated locally, albeit on a small scale (**Supplementary file 5**). Pollen evidence also suggests small-scale local cultivation

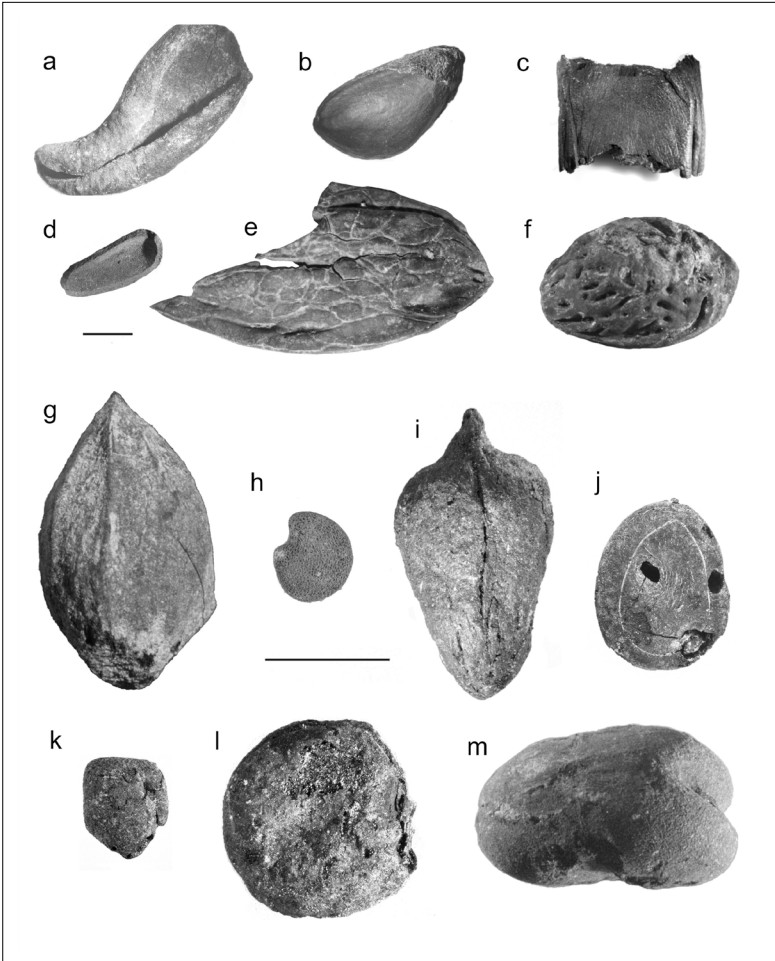

**Figure 3.** Select plant remains from the Negev Highland middens (**a**) charred almond (*Prunus amygdalus* Batsch) exocarp; (**b**) charred pistachio (*Pistacia vera* L.) drupe; (**c**) charred carob (*Ceratonia siliqua* L.) pod fragment; (**d**) uncharred stone pine (*Pinus pinea* L.) outer seed coat fragment; (**e**) uncharred walnut (*Juglans regia* L.) endocarp fragment (**f**) charred peach (*Prunus persica* [L.] Batsch) endocarp; (**g**) charred cherry/plum (*Prunus* subgen. *Cerasus/Prunus*) endocarp; (**h**) uncharred aubergine (*Solanum melongena* L.) seed; (**i**) charred jujube (*Ziziphus jujuba/mauritiana*) endocarp; (**j**) charred *Nile acacia* (*Vachellia nilotica* [L.] P.J.H.Hurter & Mabb.) seed; (**k**) charred fenugreek (*Trigonella foenum-graecum/berythea*) seed; (**l**) charred white lupine (*Lupinus albus* L.) seed; (**m**) charred broad bean (*Vicia faba* L.). Scale bars = 5 mm for both a-f and g-m; all photos in grayscale (photographed by: Daniel Fuks and Yoel Melamed). Additional photos of select plant remains appear in *Figure 3—figure supplement 1*.

The online version of this article includes the following figure supplement(s) for figure 3:

**Figure supplement 1.** Supplementary photos of select plant remains from the Negev Highland middens.

of hazel—an additional domesticate unattested in the Southern Levant before the Roman period (*Supplementary files 4 and 5*).

Another important ancient economic plant found in the assemblages is the Nile acacia (*Vachellia nilotica* (L.) P.J.H.Hurter & Mabb.), which does not grow today in the Negev. Previous archaeobotanical finds of Nile acacia in the Levant all come from Roman-period sites in the Dead Sea rift valley, which *Kislev, 1990* interpreted as a component of the ancient flora in this area marked by pockets of Sudanian vegetation. However, this was also an important region for desert-crossing camel caravan commerce, connecting Arabia, the Red Sea, and the Mediterranean. Nile acacia seed finds from Elusa (*Figure 3*) are the first of their kind from outside the phytogeographic region of Sudanian vegetation, but they remain within the ancient caravan trade routes connecting the Red Sea and the Mediterranean. Therefore, we consider Nile acacia seeds to represent a Roman-period introduction to the Levant, whether as objects of cultivation or of trade at the Negev desert route sites. Other exotic trees

commonly used for quality wood and craft were identified by pollen and/or charcoal, including: cedar of Lebanon (*Cedrus libani* A.Rich.), European ash (*Fraxinus excelsior* L.), and boxwood (*Buxus semper-virens* L.). Cedar was identified by both charcoal and pollen, suggesting local garden cultivation (see *Langgut et al., 2021* and *Supplementary files 3 and 4*).

The Early Islamic period middens were more concentrated in plant remains, and it is in them that most of the rare domesticated species, RAD crops included, were found (*Supplementary file 2*). Samples containing the unique finds of white lupine and non-indigenous jujube—which are unprecedented in Southern Levantine archaeobotany—were dated to the Umayyad or early Abbasid period (mid-7[th] – late 8[th] c. cal. CE at 2σ; see *Figure 1* and *Supplementary file 6*). However, historical studies have identified these species in Roman-period texts of the Southern Levant (*Amar, 2000*). All other RAD species found in the Negev Highlands are attested to in the Southern Levantine archaeobotanical record of the 1[st] c. BCE–4[th] c. CE (*Supplementary file 7*). The near absence of these crop species in the Negev Highland Byzantine middens compared with the Early Islamic middens is likely the result of conditions favoring deposition and preservation of archaeobotanical remains in the latter, such as a much higher concentration of apparently hearth-derived domestic waste. Therefore, we do not consider the paucity of RAD crops in the Byzantine middens to be evidence of their absence. However, one crop for which there is no pre-Islamic evidence in the Southern Levant is the aubergine. The sediment sample from Shivta containing aubergine seeds was dated to the Abbasid period (772–974 cal CE at 2σ), supporting previous finds from Abbasid Jerusalem (*Amichay et al., 2019*; *Amichay and Weiss, 2020*; *Samuel, 2001*).

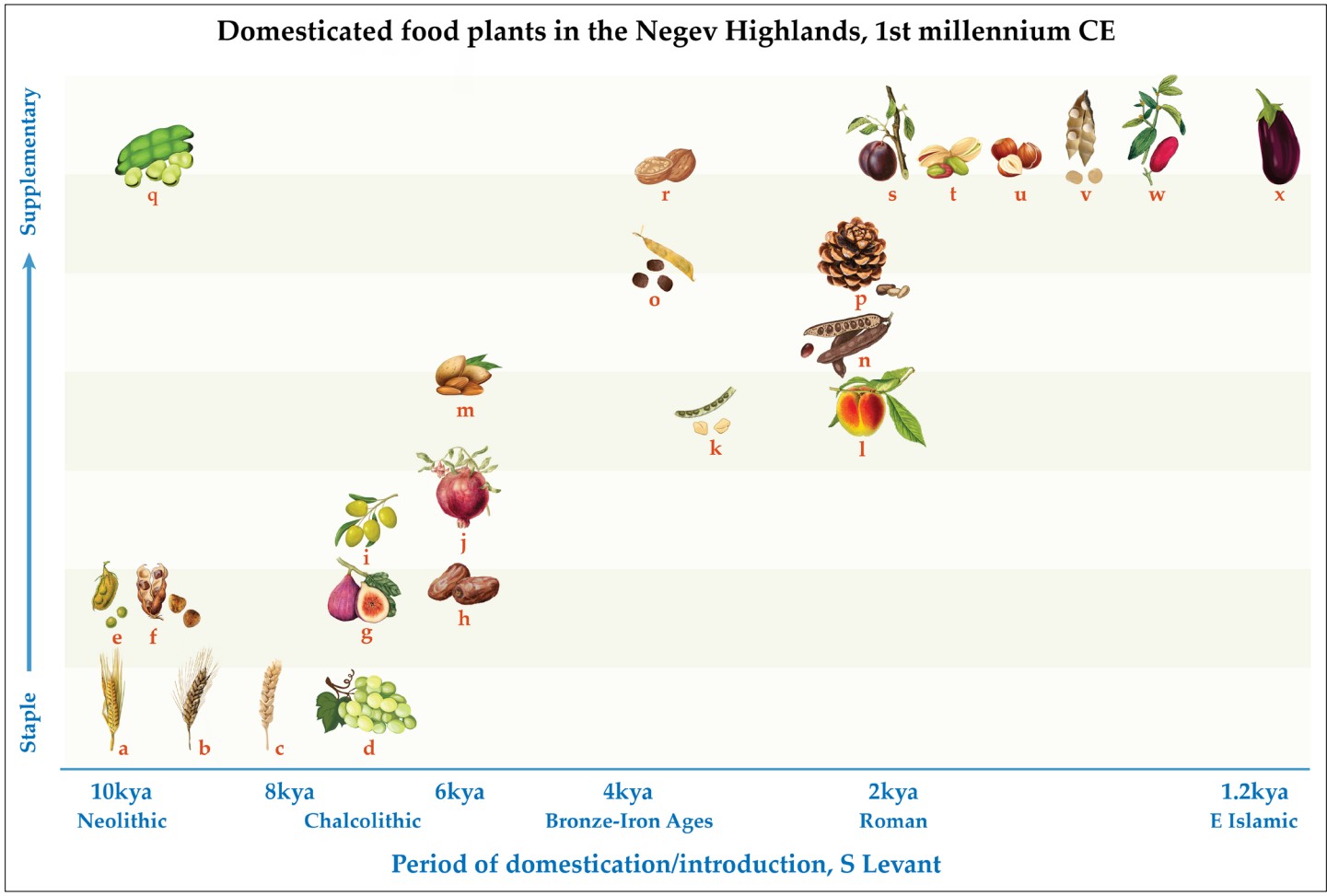

**Figure 4.** Schematic representation of domesticated food plants according to their frequency in the first millennium CE Negev Highland sites and period of initial domestication in, or introduction to, the Southern Levant: (**a**) barley, (**b**) free-threshing tetraploid wheat, (**c**) free-threshing hexaploid wheat, (**d**) grape, (**e**) lentil, (**f**) bitter vetch, (**g**) fig, (**h**) date, (**i**) olive, (**j**) pomegranate, (**k**) fenugreek, (**l**) peach, (**m**) almond, (**n**) carob, (**o**) Spanish vetchling, (**p**) stone pine, (**q**) broad bean, (**r**) walnut, (**s**) plum/cherry, (**t**) pistachio, (**u**) hazel, (**v**) white lupine, (**w**) jujube, (**x**) aubergine.

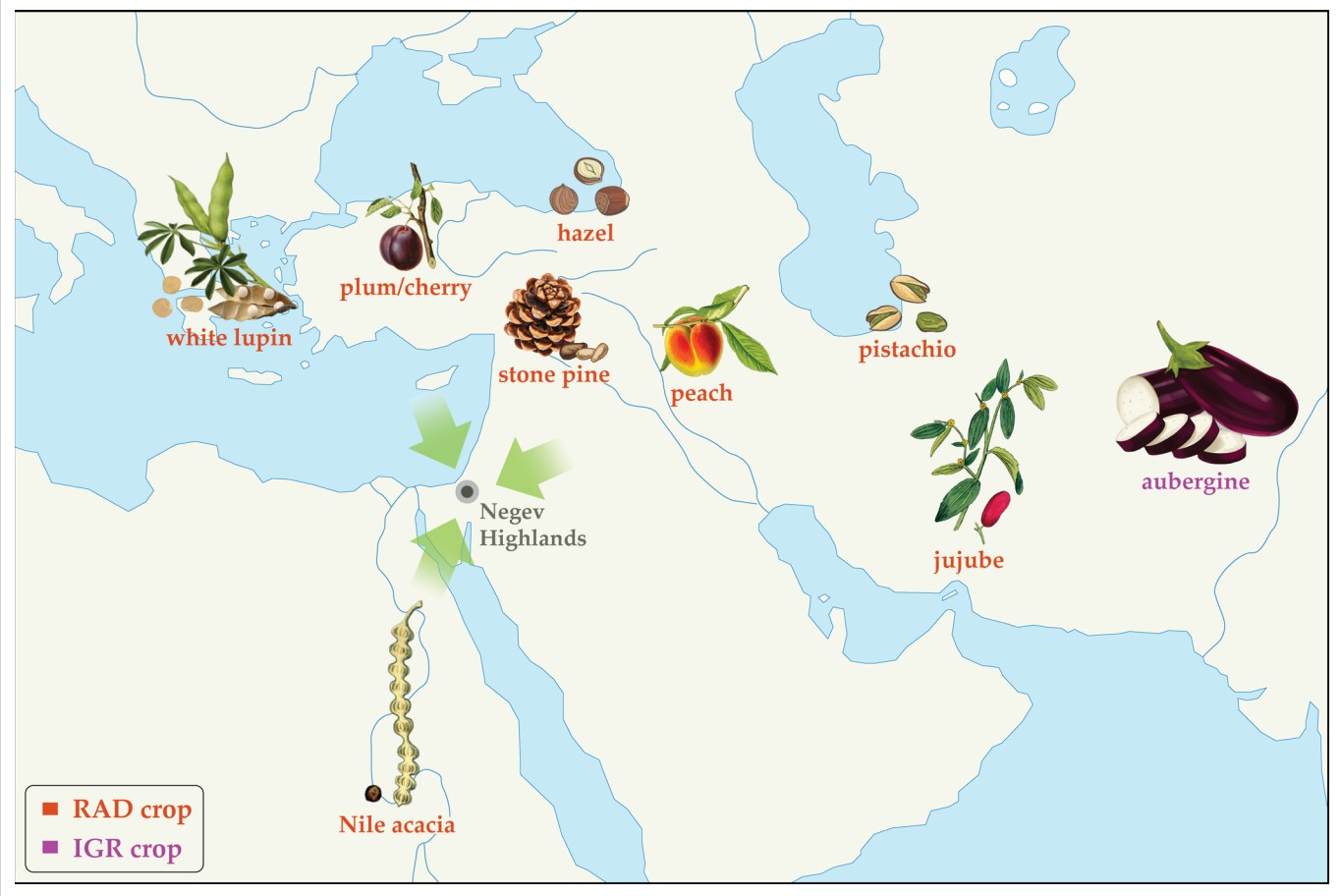

**Figure 5.** Schematic representation of directions of first millennium CE crop diffusion into the Southern Levant based on plants attested to in the Negev Highland middens. Roman Agricultural Diffusion (RAD) crops are labeled red; Islamic Green Revolution (IGR) crops are labeled purple. Placements on map convey general directions of diffusion, not necessarily precise origins.

Considering together the domestic plants evident in the Negev Highlands according to their period of first attestation in the Southern Levant, archaeobotanically and historically, offers a window onto processes of long-term crop diffusion (*Supplementary file 7*). While their quantities and ubiquities indicate that RAD and IGR crops were initially of minor significance, they make up over a third of the domesticated species found in the Negev Highland middens (*Figure 4*, *Supplementary file 7*). All the more surprising considering the Negev Highlands' desert and present-day peripheral status, this new data reveals for the first time the extent of western influence on local agriculture and trade (*Figure 5*).

## Discussion

The critical mass afforded by the new, systematically retrieved and identified plant remains from Late Antique Negev Highland trash mounds allows not only reconstructions of the local plant economy, but also insights into the long-term dispersal of crop plants. Of the Negev Highland plant remains, only aubergine is an IGR crop (*Figures 4–5*; *Supplementary file 7*). Together with seeds from Abbasid Jerusalem, seeds from the Negev Highland middens are among the earliest archaeobotanical finds of this plant in the Levant and are roughly contemporaneous with the earliest textual references to aubergine in the region (*Watson, 1983*; *Amar, 2000*). Significantly, aubergine is the only summer crop in the Negev Highlands plant assemblage. In other regions of the Southern Levant, summer crops were certainly cultivated in the Roman period (*Decker, 2009b*; *Feliks, 2008*), but the Early Islamic introduction of aubergine is consistent with Watson's claim that summer cultivation expanded in this later period (*Watson, 1983*; *Samuel, 2001*; *Van der Veen and Morales, 2011*). Ultimately, widespread adoption of summer-winter crop rotation in the Mediterranean region effected changes

in people's diets and work routines. Yet these changes clearly did not occur rapidly. To be fair, the Early Islamic assemblages from the Negev Highlands do not offer enough of a time perspective to fully gauge the effects of Early Islamic crop introduction on their own as they span only the first 200–300 years of Islam. Yet it is also possible that finds from the 7th–8th century middens reflect carry-overs from Byzantine agronomic traditions and techniques. Had crop introductions been inundating and pervasive during the Early Islamic period, we expect they would have been more apparent in Negev Highland crop diversity.

By contrast, the Negev Highlands crop basket highlights the influence of RAD, particularly on arboriculture. Roughly one-third of the domesticated food plants found in the Negev Highlands were evidently introduced to the Southern Levant between the 1st c. BCE and the 4th c. CE. Among those identified by carpological remains are pistachio, stone pine, peach, plum/cherry, jujube, and white lupine, plus carob which is a local wild species but was apparently not fully domesticated until the Hellenistic-Roman period (*Van der Veen and Hamilton-Dyer, 1998*; *Vermeeren and Cappers, 2002*; *Bouby and Marinval, 2004*; *Bosi et al., 2020*; *Figure 4*; *Supplementary file 7*). Considering pollen remains, hazel is an additional RAD species identified in the Negev Highlands by pollen remains; its pollen was also found in Herod's garden at Caesarea, probably as an imported ornamental (*Langgut, 2022*), and endocarp remains were retrieved from the Nahal Arugot cave inhabited by Bar Kokhba rebels in 135 CE (*Kislev and Simchoni, 2006*). The different RAD species were originally domesticated in various parts of the Eastern Mediterranean and Asia; a schematic sketch of the directions of first millennium CE diffusion of these crops is portrayed in *Figure 5*. Although not a food plant, we also consider Nile acacia to be a RAD crop introduced from Egypt, as noted above.

The fact that the RAD plant remains are more prevalent in the Early Islamic phase (*Supplementary file 2*) is likely the result of overall better preservation and plant richness in this phase. Therefore, we understand them to be part of the general Late Antique Negev Highlands domesticated plant assemblage, noting that their earliest secure archaeobotanical records in the Southern Levant as a whole derive mostly from the 1st c. BCE to the 2nd c. CE (*Supplementary file 7*). We acknowledge that some RAD species are first attested to at the end of the Hellenistic period of the Southern Levant in the 1st c. BCE. We nonetheless consider them RAD crops in view of chronological proximity as well as their entrenchment in local agriculture and culture during the Roman period. Allowing for gaps in the archaeobotanical record, partially compensated by textual references, it is still fair to say that the RAD plants—which comprise a significant proportion of species diversity in the Late Antique Negev Highland basket of domestic plants—were introduced to the Southern Levant over a relatively short period in Holocene history (*Figure 4*).

The snapshot presented here of the Negev Highlands' microregional crop basket supports and significantly enhances previous evidence for first millennium CE crop diffusion. Together with the archaeobotany of sites from southern Jordan (*Bouchaud et al., 2017*) and Jerusalem (*Amichay and Weiss, 2020*; *Amichay et al., 2019*), the Negev Highland plant remains attest to Roman and Byzantine dispersal in the Southern Levant of fruit crops such as peach, pear, plum, jujube, apricot, cherry, pistachio, pine nut, and hazel, among others, and to Abbasid introduction of aubergines. Altogether, this evidence suggests that RAD was a greater force in the agricultural history of the first millennium CE than the IGR, which is also the current consensus from Iberia (*Peña-Chocarro et al., 2019*). The significance of RAD is evident in the archaeobotany of additional regions, such as Italy, Northwest Europe, and Britain (*Van der Veen et al., 2008*; *Langgut, 2017*; *Bosi et al., 2020*). However, we should not dismiss the IGR on these grounds alone, since several of the proposed IGR crops are less likely to leave identifiable macroscopic traces (e.g. sugar cane, taro), and there is textual evidence for Early Islamic crop diffusion and agricultural innovation (*Amar, 2000*). Hence it may be appropriate and productive to consider RAD and IGR part of the same process of first millennium CE agricultural development, as indicated by Early Islamic expansion of Roman and Byzantine crop introductions. Clearly, the first millennium CE was an unprecedented period of change for local crop-plant species diversity in the Eastern Mediterranean and beyond. The multi-regional evidence suggests that the multi-empire combination of Roman-Byzantine and Umayyad-Abassid regimes was a major force for crop diffusion, while a likely role for developments in the Sassanid empire is underrepresented in current research. Yet the evidence presented here demonstrates that even the combined forces underlying first millennium CE crop diffusion affected, but did not immediately transform, people's diets. At least until the end of that millennium, inhabitants of the Levant and Eastern Mediterranean region continued to rely

primarily on long-tried and tested Neolithic founder crops and early fruit domesticates. Indeed, this situation widely persisted until the latter second millennium CE.

In conclusion, the new microregional data presented above supports an emerging multi-regional picture of both an unprecedented period for plant migrations and food diversity in the first millennium CE as well as gradual and incomplete local adoption. This is evident from Late Antique Negev Highland archaeobotanical assemblages within which plants first attested to in the Southern Levant during this period account for one-third of the domesticated plant species diversity—more than any other period represented in the assemblage (*Figure 4*). Among these crops, only the aubergine was an Early Islamic introduction, suggesting that Roman Agricultural Diffusion (RAD) was a greater force for the intercontinental movement of crop plants than the proposed Islamic Green Revolution (IGR). However, specimen counts and ubiquity of both RAD and IGR plant species are very low in the Negev Highland assemblages, indicating slow incorporation into local foodways and agriculture. These findings present a window onto a wider perspective on the last 10+ millennia of Southwest Asian crop diffusion, in which the first millennium CE is unprecedented for the diversity of plant species in motion yet consistent with a long-term pattern of gradual local adoption.

## Materials and methods
### Sampling and screening

Eleven middens from the three sites, Elusa, Shivta, and Nessana, were excavated at approximately 10 cm height intervals to ensure chronological control (*Figure 1*). Loci and baskets were assigned by a combination of stratigraphy and sediment features during excavation. A three-pronged sifting strategy was adopted to maximize retrieval of artifacts and organic remains, while enabling complementary resolutions of analysis: (1) Most excavated sediment was dry screened on-site through 5 mm sieves. (2) Wet screening through 1 mm mesh was performed on two buckets (~20 l) from each excavated locus-basket. (3) One additional bucket from each locus-basket was set aside for fine screening. For ease of reference, (1) and (2) above are collectively referred to as *course-sift samples*, and (3) is referred to as *fine-sift samples*.

Due to the high volume of samples and the extremely high concentration of seeds within them, a subsampling strategy based on sieve mesh size was adopted for the fine-sift samples. Selected buckets of sample sediment were divided into 3 liter subsamples which were processed by flotation or fine-mesh dry screening, and sieved using graduated sieves at 4 mm, 2 mm, 1 mm, 0.5 mm, and sometimes 0.3 mm mesh sizes. One additional source of identified seeds was an assemblage of dissected charred dung pellets from two of the middens (*Dunseth et al., 2019*).

All flotation light fraction and heavy residue was sorted at the ≥2 mm mesh size. Light fraction was studied at 1 mm and 0.5 mm mesh sizes for select samples, such that at least three 1 mm samples and one 0.5 mm sample (6cc of 0.5 mm light fraction) were sorted for each period on each site. Fine-sift samples were sorted using an Olympus SZX9 stereo microscope. Course-sift samples were sorted by volunteers and archaeology students during the excavation and thereafter. Seed finds from the course sifting were visually examined with the aid of a stereo microscope and rare specimens were taken to the Bar-Ilan University Archaeobotany Lab for identification.

For palynological analysis, sediment samples from the middens were collected, but these were all pollen barren, probably because of oxidation. Pollen from the reservoir and the northern church at Shivta contributed additional taxa, as did wood and charcoal analyses. Results of pollen and wood analyses were previously published by Langgut et al. (*Langgut et al., 2021*; *Bar-Oz et al., 2019*) and are summarized in *Supplementary files 3–5*. Information on previous archaeobotanical records of cultivated species was retrieved from the cited literature and lab records, as well as from online databases of archaeobotanical finds (*Kroll, 2005*; *Riehl and Kümmel, 2005*; *Núñez et al., 2011*).

### Chronology

The excavations' stratigraphic, ceramic, and radiocarbon analyses enabled the differentiation of five chronological phases obtained from the middens (*Bar-Oz et al., 2019*; *Fuks et al., 2020b*): Roman (ca. 0–300 CE), Early Byzantine (ca. 300–450 CE), Middle Byzantine (ca. 450–550 CE), Late Byzantine (ca. 550–650 CE) and Umayyad (ca. 650–750 CE), which was adjusted slightly based on radiocarbon dates presented herein. This enabled the detection of trends within the Byzantine period as well as

broader chronological comparisons. These periods are each represented by between one and four middens, and some middens span two periods (see *Table 4*). Grouping the seed/fruit crop remains into broad periods of introduction to the Southern Levant was used to provide a general sketch of crop diffusion's local influence in time. Additional radiocarbon dates were attained for loci-baskets containing unprecedented finds for Southern Levantine archaeobotany, as well as the locus-basket containing well-preserved aubergine seeds in Shivta. The aubergine, lupine, and jujube seeds were too rare to sacrifice for direct radiocarbon dating so barley grains were selected from the very same sediment sample within each locus-basket and dated at the Poznan Radiocarbon Laboratory (*Supplementary file 6*).

## Plant remain identifications

All identifications of carpological remains were made with reference to the Israel National Collection of Plant Seeds and Fruits at Bar-Ilan University. Cereal grain morphometry was employed to identify candidates, using the Computerized Key of Grass Grains developed by Mordechai Kislev's laboratory (*Kislev et al., 1995*; *Kislev et al., 1997*; *Kislev et al., 1999*). As aids to identification and analysis, local plant guides were consulted, particularly the *Flora Palaestina* (*Zohary and Feinbrun-Dotan, 1986*). Additional floras of Mediterranean, Irano-Turanian, and Saharo-Arabian phytogeographic regions were consulted as needed (*Townsend and Guest, 1966*; *Davis, 1966*; *Meikle, 1977*; *Zohary et al., 1980*; *Feinbrun-Dothan and Danin, 1991*; *Turland et al., 1995*; *Boulos, 1999*; *Danin, 2004*). Identification criteria for rare, domesticated plant specimens discussed in the main text are summarized below (see also *Figure 3*):

### Aubergine (*Solanum melongena* L.)

*S. melongena* and other *Solanum* seeds are laterally compressed, broadly oval-shaped, and under 5 mm in maximal length. *S. melongena* seeds are distinguished from wild *Solanum* seeds of the Southern Levant by their larger size, reticulated seed coat pattern, and the wide ovoid hilum set in a recess in the seed's lateral outline (*Van der Veen and Morales, 2011*; *Amichay and Weiss, 2020*). This distinction includes *S. incanum* L. which was identified at Byzantine Ein Gedi and is considered by some to be the wild progenitor of *S. melongena* (*Melamed and Kislev, 2005*). The latter two criteria also distinguish *S. melongena* from domesticated *Capsicum* spp. Based on these criteria, we identified three definitive *S. melongena* seeds from Umayyad Shivta (Area E, Locus 504, Basket 5029). Poor preservation precludes definitive identification for an additional three fragmented seeds from Umayyad Nessana (Locus 102) for which *S. melongena* nonetheless appears to be the only candidate (*Figure 3—figure supplement 1*). All of the above were preserved uncharred.

### Cherry/plum (*Prunus* subgen. *Cerasus/Prunus*)

A single charred ovoid endocarp with a pointed apex, elliptical base (5 mm by 2.5 mm), and smooth surface was found in a course-sift sample from Umayyad Shivta (Area K1, Locus 165, Basket 1652; *Figure 3—figure supplement 1*). Its length from apex to base is 12.67 mm, width 9.33 mm, and breadth 7.67 mm. A ventral ridge runs down the length of the endocarp, from apex to base, accompanied by two ridges on either side and at equal distance from the central ridge. However, the right ventral ridge appears only on the top third of the endocarp while the left ventral ridge is visible in the top two-thirds. The dorsal side is marked by a single longitudinal ridge. The above characteristics ruled out apricot, peach, and almond, and leave cherry and plum as candidates (*Prunus* subgen. *Cerasus/Prunus*). Due to the wide variety of plum and cherry cultivars (*Depypere et al., 2007*) which were not fully covered by the reference collection used, we did not identify to species.

### Nile acacia (*Vachellia nilotica* (L.) P.J.H.Hurter & Mabb.)

*Vachellia* (syn. *Acacia*) is a genus in the Mimosoideae subfamily of the Fabaceae. Seeds of Mimosoideae species native to the Southern Levant are elliptical to ovate and compressed. On each face of the seedcoat, a conspicuous pleurogram delimits an ovate areole (*Gunn, 1984*; *Al-Gohary and Mohamed, 2007*). The pleurogram may either be open-ended and U-shaped/horseshoe-shaped, or closed and concentric to the seed contour. To identify seeds with these traits found in the middens, we compared seeds of Mimosoideae species native to the Southern Levant, based on samples in the Israel National Collection of Plant Seeds and Fruits: (i) *Vachellia nilotica* (L.) P.J.H.Hurter & Mabb. syn.

*Acacia nilotica* (L.) Willd. ex Delile; (ii) *Senegalia laeta* (R.Br. ex Benth.) Seigler & Ebinger syn. *Acacia laeta* R.Br. ex Benth.; (iii) *Acacia pachyceras* O. Schwartz; (iv) *Vachellia tortilis* subsp. *raddiana* (Savi) Kyal. & Boatwr. syn. *Acacia raddiana* Savi; (v) *Vachellia tortilis* (Forssk.) Galasso & Banfi syn. *Acacia tortilis* (Forssk.) Hayne; (vi) *Faidherbia albida* (Delile) A.Chev.; and (vii) *Prosopis farcta* (Banks & Sol.) J.F.Macbr. We observed that *V. nilotica* seeds are distinguished by the following characteristics:

1. The pleurogram's border (linea fissura) is closed, creating an ovate areole (*Figure 3–figure supplement 1*).
2. The areole is largest, relative to seed size, in *V. nilotica*, i.e., the distance from the linea fissura to the seed edge is shortest in this species (*Supplementary file 8*).
3. The areole's widest part is in the top third of the seed (*Supplementary file 8*).
4. A protrusion is present next to the hilum which we observed to be unique to *V. nilotica* seeds among the above species.

*V. nilotica* seeds tend to be the largest of the above except for *P. farcta*, although interspecies diversity leads to size overlap between *V. nilotica*, *A. pachyceras* and *V. tortilis* subsp. *raddiana*. *P. farcta* seeds are like *Vachellia* spp. seeds in shape but tend to be larger than most *Vachellia* seeds and more ovate to pear-shaped. Their pleurograms are visibly open. Charred *V. nilotica* seeds were identified using a combination of criteria (1)-(4) above in midden samples from Elusa (Area A1, Locus 1/10a; A4, L. 4/06 a-4/07 a; *Figure 3—figure supplement 1*). Remains of *Vachellia* were identified also in other Negev Highland sites: One seed from Nessana (A, L. 125, B. 1446) was identified as *Vachellia* sp., while a single seed from Shivta (K1, L. 153, B. 1579) could only be identified as *Vachellia/Prosopis farcta* due to poor preservation.

### White lupine (*Lupinus albus* L.)

Three species of lupine (*Lupinus*) which grow today in the Southern Levant are distinct for their large (ca. 1 cm), compressed quadrangular seeds: *L. palaestinus* Boiss., *L. pilosus* L., and the cultivated *L. albus* L. Viewed laterally, the seeds of these species have a near-circular, or D-shaped outline and, frequently, a visible depression or dimple. The triangular radicle forms the perimeter's straightest side, while the hilum leads from the radicle tip toward the lens at an angle such that the lens and radicle are on perpendicular sides with the hilum cutting across between the two. The lens is nearly as large as the hilum and both are elliptic. The seed coat surrounds the hilum by a characteristic elliptical protrusion. As is common among domesticated legumes in general, the seed coat of cultivated *L. albus* is much thinner than its local wild relatives. *L. albus* also has a much smoother outer seed coat than the highly tuberculate seed coats of *L. palaestinus* and *L. pilosus*. The *L. albus* seed coat consists of at least two layers visibly distinct in cross-section, with the outer layer having a smooth surface and the inner layer having a grainy surface. An additional feature distinguishing *L. albus* seeds from *L. palaestinus/pilosus* is the presence of a clear transverse ridge separating the radicle depression and the hilum on the seed surface. In *L. palaestinus/pilosus*, by contrast, the radicle depression and hilum are essentially contiguous, running smoothly one into the other.

Three candidates for lupine seeds were identified among course-sifted charred archaeobotanical remains from Nessana (Area A, Locus 101, Baskets 1008/1 and 1040/2). The single seed from Basket 1040 (*Figure 3—figure supplement 1*) is compressed with a lateral depression and a near-circular quadrangle in outline measuring 7.0 × 7.5 mm. Remains of a triangular radicle on the seed's straight side are clearly visible. These features narrowed its identification to one of the three aforementioned *Lupinus* species. Both lens and hilum are visible; their shape and orientation match those of *Lupinus* seeds. A slight but clear protrusion separating the hilum from the radicle depression warrants identification as *Lupinus albus*. Remnants of a thin and grainy seed coat are visible in the center of the cotyledon's surface, in the middle of the lateral depression, consistent with *L. albus*.

Two additional seeds from Basket 1008/1 show characteristic lupine (*Lupinus* sp.) hila and radicles. The seeds measure 6.5 × 7.0 mm and 7.5 × 8.0 mm which, together with their D-shaped outlines, corresponds with that typical to the large lenticular lupine species mentioned above. The two seeds from Basket 1008/1 are broader than the *L. albus* seed from Basket 1040/2, and the characteristic lateral depression is not visible. This is apparently due to lateral swelling and partial disfiguration during charring as is common in charred legume seeds. In the larger of the two seeds, a thin, grainy seed coat is visible surrounding the triangular radicle and covering one of the cotyledons. In that

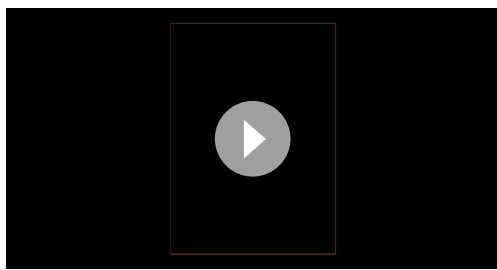

**Video 1.** Micro-CT longitudinal scans of *Z. jujuba/mauritiana* endocarp.
https://elifesciences.org/articles/85118/figures#video1

same seed, a topographic separation between the radicle depression and hilum justifies identification as *L. albus.*

## Jujube (*Ziziphus jujuba/mauritiana*)

A single charred obconical-mucronate endocarp was found from Umayyad-period layers at Shivta (Area E, Locus 501, Basket 5108). Micro-CT scanning was conducted using a Bruker desktop SkyScan 1174 at the Laboratory of Bone Biomechanics, Hebrew University of Jerusalem (optical resolution: 9.6 μm; exposure: 4500ms; rotation step: 0.400 degrees; 180-degree rotation option was used; 0.25 mm thick aluminum filter). The scans demonstrated the specimen to be spherically hollow with remnants of a partition (*Videos 1–2*), confirming its status as a fruit endocarp. The external endocarp dimensions (11.16 mm × 6.0 mm × 5.33 mm) and its obconical with markedly narrowing apex (*Figure 3—figure supplement 1*) are unique to certain varieties of *Ziziphus jujuba/mauritiana*. The specimen's pointed edges tapered slightly and the external grooves characteristic of *Z. jujuba/mauritiana* are barely recognizable, apparently the result of abrasion during or following charring. Remnants of the characteristic v-shaped basal scar between the two endocarp halves (*Jiang et al., 2013*, their Figure 6) are barely visible, again likely due to abrasion. Species with similar endocarps include local wild types of *Ziziphus* (*Z. spina-christi* (L.) Willd., *Z. lotus* Lam., *Z. nummularia* (Burm.f.) Wight & Arn.), but their endocarps are always spherical and never obconical-mucronate to the extent of *Z. jujuba/mauritiana* and the specimen at hand.

## Spanish vetchling (*Lathyrus clymenum* L.)

Identification of *Lathyrus clymenum* L. was based on morphological similarity to ancient *L. clymenum* seeds identified from Tel Nami by *Kislev et al., 1993*. Diagrams and measurements reported by *Sarpaki and Jones, 1990* for a large number of *L. clymenum* seeds from Late Bronze Age Akrotiri and Knossos were also used.

The following generalized description refers to the identified *L. clymenum* seeds from Shivta and Nessana: The seeds are laterally compressed, and nearly rectangular in circumstance. In lateral view, the radicle lies on the short side, perpendicular to the long side where the hilum lies (*Figure 3—figure supplement 1*). The radicle forms a somewhat planar face, especially in comparison with the other sides of the seed. The dorsal side (parallel to that on which the hilum lies), is conspicuously carinated, whereas the ventral side is only moderately carinated. The hilum occupies over half the length of the ventral side. It begins at one end of the ventral side (near the radicle) and ends just before the circular lens. The thin seed coat is neither perfectly smooth nor tuberculate but appears grainy at a magnification of ca. 40 X.

Charred *L. clymenum* seeds were identified at Nessana, Area A (Locus 106, Basket 1255 cf. L. 106, B. 1257; L. 101, B. 1032) and several from Area K at Shivta (L. 153, B. 1588,1610; L. 158, B. 1618; L. 166, B. 1658; L. 169, B. 1678,1703; L. 172, B. 1689). The positions, shapes, and relative sizes of the hilum and lens matched those of the Tel Nami *L. clymenum* seeds and the depictions of *Sarpaki and Jones, 1990*. The same is true for seed coat thickness and texture, as well as the markedly carinated dorsal side. One seed from Shivta (K1, L. 153, B. 1588) measured below the range of Tel Nami seed dimensions (*Supplementary file 9*). However, its relative dimensions and clear morphology justified unequivocal identification as *L. clymenum.*

## Acknowledgements

As part of a Ph.D. dissertation conducted at Bar-Ilan University, this research was supported by

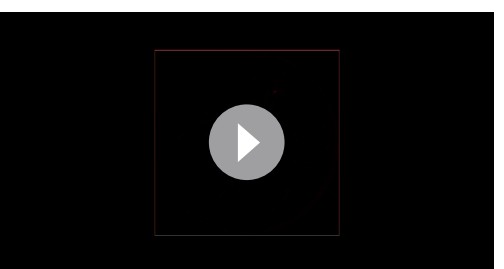

**Video 2.** Micro-CT lateral scans of *Z. jujuba/mauritiana* endocarp.
https://elifesciences.org/articles/85118/figures#video2

the Bar-Ilan Doctoral Fellowships of Excellence Program, the Rottenstreich Fellowship of the Israel Council for Higher Education, and the Molcho Fund for agricultural research in the Negev awarded to D.F. As part of the NEGEVBYZ project, this research was also supported by the European Research Council under the European Union's Horizon 2020 Research and Innovation Programme (NEGEVBYZ project, grant no. 648427) and the Israel Science Foundation (grant no. 340–14) awarded to G.B.O. Manuscript preparation was further supported by a Newton International Fellowship of the British Academy (NIF23/100633) and a Marie S. Curie International Fellowship of the European Commission's Horizon 2020 Framework Programme (CroProLITE project, grant no. 101025677), awarded to D.F. Archaeology was conducted on behalf of the Zinman Institute of Archaeology, University of Haifa, under licenses of the Israel Antiquities Authority (Elusa: G-69/2014, G-10/2015, G-6/2017; Shivta: G-87/2015, G-4/2016; Nessana: G-4/2017). We also wish to thank the Israel Nature and Parks Authority for facilitating the excavations at Elusa, Shivta, and Nessana, as well as Ami and Dina Oach of Shivta Farm. For assistance with processing during the excavations we are grateful to Ifat Shapira, Uri Yehuda, Ruti Roche, Gabriel Fuks, Aehab Asad, Ari Levy, Yaniv Sfez, and countless other volunteers. We also wish to thank Yael Mahler-Slasky, Tammy Friedman, Anat Hartmann-Shenkman, Michal David, Suembikya Frumin, Noam Even, Itamar Berko, and Oriya Bashari for laboratory assistance; Senthil Ram Prabhu Thangadurai and Ron Shahar of the Hebrew University of Jerusalem's Laboratory of Bone Biomechanics for micro-CT scanning; and Sapir Haad for graphics.

## Additional information

### Funding

| Funder | Grant reference number | Author |
|---|---|---|
| Bar-Ilan University | Doctoral Fellowships of Excellence Program | Daniel Fuks |
| Bar-Ilan University, Israel Council for Higher Education | Rottenstreich Fellowship | Daniel Fuks |
| Molcho Fund | Agricultural Research in the Negev | Daniel Fuks |
| Horizon 2020 Framework Programme | European Research Council Grant agreement 648427 | Guy Bar-Oz |
| Israel Science Foundation | Grant 340-14 | Guy Bar-Oz |
| British Academy | NIF23/100633 | Daniel Fuks |
| University of Cambridge, Horizon 2020 Framework Programme | Marie S. Curie International Fellowship 101025677 | Daniel Fuks |

The funders had no role in study design, data collection and interpretation, or the decision to submit the work for publication.

### Author contributions

Daniel Fuks, Conceptualization, Data curation, Formal analysis, Funding acquisition, Investigation, Methodology, Writing – original draft, Writing – review and editing; Yoel Melamed, Supervision, Validation, Methodology, Writing – review and editing; Dafna Langgut, Data curation, Methodology, Writing – review and editing; Tali Erickson-Gini, Yotam Tepper, Methodology, Writing – review and editing; Guy Bar-Oz, Resources, Supervision, Funding acquisition, Methodology, Project administration, Writing – review and editing; Ehud Weiss, Resources, Supervision, Validation, Methodology, Writing – review and editing

### Author ORCIDs

Daniel Fuks ⓘ https://orcid.org/0000-0003-4686-6128
Yoel Melamed ⓘ https://orcid.org/0000-0003-0952-708X

Dafna Langgut [ORCID] http://orcid.org/0000-0002-4824-1044
Tali Erickson-Gini [ORCID] http://orcid.org/0000-0003-2902-3831
Guy Bar-Oz [ORCID] https://orcid.org/0000-0002-1009-5619
Ehud Weiss [ORCID] https://orcid.org/0000-0002-9730-4726

**Decision letter and Author response**
Decision letter https://doi.org/10.7554/eLife.85118.sa1
Author response https://doi.org/10.7554/eLife.85118.sa2

## Additional files

### Supplementary files
- MDAR checklist
- Supplementary file 1. Carpological plant remains from Negev Highland middens.
- Supplementary file 2. Presence/absence of domesticated species in Negev Highland middens by period (carpological remains).
- Supplementary file 3. Identified wood and charcoal taxa from Shivta, Nessana, and Elusa.
- Supplementary file 4. Identified pollen from Shivta reservoirs and garden.
- Supplementary file 5. Combined evidence for fruit/nut trees.
- Supplementary file 6. Radiocarbon dating of select loci.
- Supplementary file 7. Earliest archaeobotanical evidence in the S Levant for domestication/introduction of Negev Highland domesticated plants.
- Supplementary file 8. Some *Acacia* spp. seed measurements from the Israel National Collection of Plant Seeds and Fruits.
- Supplementary file 9. Select *L. clymenum* seed measurements from Tel Nami.

### Data availability
Only identified plant taxa are reported in the results of this study and all relevant data are included in the manuscript and supplementary materials. Source data may be found in *Table 4—source data 1*, *Table 4—source data 2* and *Table 4—source data 3*.

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
