## [Editor Report]

The study presents important findings on the timing and movement of crops in the Near East. The authors provide convincing data supporting a predominant contribution of Roman Agricultural Diffusion to the spread of a number of cultigens in the region. The work will be of interest to those thinking about the timing and movement of the diffusion of agricultural crops post-domestication.

---

## [Decision Letter]

**Decision letter after peer review:**

Thank you for submitting your article "Unprecedented yet gradual nature of first millennium CE intercontinental crop plant dispersal revealed in ancient Negev desert refuse" for consideration by *eLife*. I would like to apologize for the lengthy review process; it turned out to be quite difficult to find reviewers! In the end, your article has been reviewed by 2 peer reviewers, one of whom is a member of our Board of Reviewing Editors, and the evaluation has been overseen by Detlef Weigel as the Senior Editor. Both reviewers were very positive about the work and I share their enthusiasm. The following individual involved in the review of your submission has agreed to reveal their identity: Simcha Lev-Yadun (Reviewer #1).

Essential revisions:

The reviewers and I agree this is nice work that can make for an excellent contribution to the field. Both reviewers have a series of relatively minor suggestions. The two main points I would like you to address in reviews:

1) Reviewer #2 had a question about the interpretation of missing RAD crops in the record and how that should be interpreted.

2) The reviewers disagreed about the presentation. Reviewer 2 found the tables and presentation of the data quite difficult to follow and offered a number of suggestions about turning the tables into figures. Reviewer 1 very much liked the tables. I suggest you keep the tables, but perhaps move them to supplement where dedicated readers can still find the information. I think some consideration of making the presentation easier for non-expert reviewers is well worth considering.

---

## [Author Response]

Essential revisions:The two main points I would like you to address in reviews:1) Reviewer #2 had a question about the interpretation of missing RAD crops in the record and how that should be interpreted.

We agree that this is an important point and we address it through additional explanatory text (details below).

2) The reviewers disagreed about the presentation. Reviewer 2 found the tables and presentation of the data quite difficult to follow and offered a number of suggestions about turning the tables into figures. Reviewer 1 very much liked the tables. I suggest you keep the tables, but perhaps move them to supplement where dedicated readers can still find the information. I think some consideration of making the presentation easier for non-expert reviewers is well worth considering.

I believe we have found the way to adopt both reviewers’ suggestions on this point and make the presentation of data more accessible: We adopted Reviewer 1’s suggestion to add what had been Supplementary Tables 1-3 into the main text, and also listed examples of the specific crops in motion in the text. We also adopted Reviewer 2’s suggestions and made all other tables supplementary – with the exception of the current Table 4, which is essential. In addition, we replaced what had been a pie chart in Figure 4 with a new figure which offers a schematic representation of the first millennium CE Negev Highlands crop basket according to frequency and ubiquity in the archaeobotanical assemblage as well as period of first introduction/domestication in the Southern Levant. This summarizes the main findings of the paper, and also complements Figure 5, which shows the general directions of diffusion for the RAD crops found in the Negev Highlands. Throughout the piece, we also made minor edits and slight restructuring, aimed toward more precise and accessible presentation.